# Identifying subgroups of high-need, high-cost, chronically ill patients in primary care: A latent class analysis

Rowan G. M. Smeets<sub></sub>*, Arianne M. J. Elissen, Mariëlle E. A. L. Kroese, Niels Hameleers, Dirk Ruwaard

Department of Health Services Research, Maastricht University, Faculty of Health, Medicine and Life Sciences, Care and Public Health Research Institute (CAPHRI), Maastricht, The Netherlands

* rowan.smeets@maastrichtuniversity.nl

**Data Availability Statement:** Results are based on calculations by the department of Health Services Research, Maastricht University, the Netherlands, using non-public microdata from Statistics

## Abstract

### Introduction

Segmentation of the high-need, high-cost (HNHC) population is required for reorganizing care to accommodate person-centered, integrated care delivery. Therefore, we aimed to identify and characterize relevant subgroups of the HNHC population in primary care by using demographic, biomedical, and socioeconomic patient characteristics.

### Methods

This was a retrospective cohort study within a Dutch primary care group, with a follow-up period from September 1, 2014 to August 31, 2017. Chronically ill patients were included in the HNHC population if they belonged to the top 10% of care utilizers and/or suffered from multimorbidity and had an above-average care utilization. In a latent class analysis, forty-one patient characteristics were initially used as potential indicators of heterogeneity in HNHC patients' needs.

### Results

Patient data from 12 602 HNHC patients was used. A 4-class model was considered statistically and clinically superior. The classes were named according to the characteristics that were most dominantly present and distinctive between the classes (i.e. mainly age, household position, and source of income). Class 1 ('older adults living with partner') included 39.3% of patients, class 2 ('older adults living alone') included 25.5% of patients, class 3 ('middle-aged, employed adults with family') included 23.3% of patients, and class 4 ('middle-aged adults with social welfare dependency') included 11.9% of patients. Diabetes was the most common condition in all classes; the second most prevalent condition differed between osteoarthritis in class 1 (21.7%) and 2 (23.8%), asthma in class 3 (25.3%), and mood disorders in class 4 (23.1%). Furthermore, while general practitioner (GP) care utilization increased during the follow-up period in the classes of older adults, it remained relatively stable in the middle-aged classes.

Netherlands and claims data from health care information center 'Vektis'. Under certain conditions, these microdata are accessible for statistical and scientific research. For further information and access queries: (microdata@cbs. nl). In addition, data from general practices connected to primary care group 'HZD' were used for the analysis. The data from the general practices are owned by the general practices and 'HZD'; requests for access to this data should be submitted to 'HZD' (info@hzd.nu)

**Funding:** The current study was funded by primary care group 'HZD', the Netherlands (https://www. hzd.nu). The funding agency had no role in study design, data collection and analysis, decision to publish, or preparation of the manuscript.

**Competing interests:** AE is currently serving as an Academic Editor for PLOS ONE. Furthermore, our funding source (i.e. primary care group 'HZD') is a commercial source. This does not alter our adherence to PLOS ONE policies on sharing data and materials.

## Conclusions

Although the HNHC population is heterogeneous, distinct subgroups with relatively homogeneous patterns of mainly demographic and socioeconomic characteristics can be identified. This calls for tailoring care and increased attention for social determinants of health.

## Introduction

Due to increasing numbers of chronically ill patients, in particular with multimorbidity, and rising health care costs, Western health care systems are faced with challenges to deliver high-quality, person-centered, and sustainable care [1–3]. In response to these developments, accountable care organizations (ACOs) were introduced in the United States several years ago [4–6]. Within ACOs, a value-based payment system is designed to incentivize providers to share accountability for the quality and cost of care for a defined population [6–8]. Likewise, more than a decade ago, 'care groups' were first introduced in Dutch primary care. In line with the ACOs, care groups unite providers, mostly general practitioners (GPs), with shared responsibility for all assigned patients receiving care for a specific chronic condition from a value-based bundled payment approach [5, 9]. These initiatives show that, similar to the US, the Netherlands aims to achieve more value-based care.

If health systems aim to increase the value of delivered care, it is crucial to focus on the population with the highest care use as they offer the largest potential for achieving improved value [10, 11]. This population with a disproportionately high care use is also referred to as the high-need, high-cost (HNHC) population [10, 12]. The identification of the HNHC population, as a subgroup of the total population, is embedded in the approach of population segmentation, which is defined as the division of a specific population into homogeneous subgroups with distinct needs and (health) characteristics [13–15]. A closely related concept in which principles of segmentation are applied, pertains to the concept of 'population (health) management' (PM),[16] as a way to promote 'population health' [17, 18]. Within population health, the focus is on the health outcomes of subgroups rather than individuals, by taking into account a large variety of determinants of health (i.e. physical, mental, social) [17, 18]. PM strategies generally aim to improve health needs of defined subgroups along 'the continuum of health and well-being', and aim to integrate services across multiple domains [16]. As such, PM strategies can be used to tailor interventions to the care needs of specific subgroups of patients, which is assumed to lead towards improving individual patients', and providers' experiences as well as population outcomes and cost (Quadruple Aim[19]).

With the growing availability of digital patient data, studies have identified common biomedical characteristics of the HNHC population, such as the high prevalence of (co-occurring) chronic conditions and mental illness [20, 21]. At the same time, studies have suggested that the HNHC population is diverse, not only in terms of patients' biomedical but also in their demographic and socioeconomic profiles [10, 20, 21]. These findings underline the importance of social determinants of health within the HNHC population. Yet, population segmentation studies have predominantly focused on specific populations, such as older adults [22–24] and Medicaid beneficiaries [25], and mainly characterized the identified patient subgroups by their biomedical characteristics (i.e., chronic diagnoses) [22–27]. Therefore, the main aim of this study was to identify and characterize, by means of latent class analysis (LCA), clinically relevant subgroups of the HNHC population in primary care, defined by demographic, biomedical, and socioeconomic patient characteristics as well as care utilization.

## Materials and methods

### Setting

This retrospective cohort study was conducted at a (primary) care group in the northern region of the Netherlands, covering 130 general practices. This care group was founded in 2009 and currently has bundled payment contracts with health insurers for the delivery of several disease management programs, including for patients with type 2 diabetes mellitus, COPD, and cardiovascular risks.

As this study used retrospective data and did not intervene into people's life or impose rules, no formal ethical approval was required (project number 164111), in line with the Dutch Medical Research (Human Subjects) Act.

### Data sources

All general practices connected to the care group were invited to extract and provide individual-level patient data from their electronic health records (EHRs). The EHR data covered 4.5 years: baseline was on September 1, 2014; the follow-up period covered three years (from September 1, 2014 to August 31, 2017). Furthermore, the EHR data were linked on the individual patient level to socioeconomic data (e.g., source of income) and health care claims data (e.g., pharmaceutical costs). Socioeconomic data were retrieved from Statistics Netherlands, which is involved in the collection, preparation, and publication of statistics on behalf of the Dutch government, science and commercial sector [28]. Claims data were retrieved from the health care information center 'Vektis', which collects and manages all claims under the Dutch Healthcare Insurance Act [29]. To ensure data confidentiality and safety, a third trusted party was involved in the provision of a pseudonymized version of the data set to the researchers.

### Participants

We selected a cohort of chronically ill patients, limited to those with a full EHR registration over the 4.5-year research period. Patients were considered chronically ill if they had registered at least one GP consultation in the 1.5 years before baseline related to one of 28 conditions defined as chronic (see Table 1) [30, 31]. Chronically ill patients were included in the HNHC population if they belonged to the top 10% of care utilizers (over follow-up period) and/or suffered from multimorbidity and had an above-average care utilization (over follow-up period). The first criterion was applied as this is one of the commonly used thresholds for identifying HNHC patients according to previous studies [20, 32, 33]. The second criterion was applied because multimorbidity brings along a challenging complexity to the organization of care, especially in light of the current single-disease management programs for single chronic conditions [2, 3]. Furthermore, care utilization was measured as the total number of GP consultations weighted by the required time investment per type of consultation (i.e. 0.5 for telephone or e-mail consultation, 1.0 for regular consultation, 2.0 for extended regular consultation, 1.5 for home visit, 2.5 for extended home visit), determined by the Netherlands Institute for Health Services Research [34]. As the weighting factors based on time investment are related to costs [35], the patients selected for this study can be considered high-need, high-cost in primary care.

### Variables

Forty-one patient characteristics were initially used as potential indicators of heterogeneity in HNHC patients' needs in the LCA. These characteristics were included based on scientific studies describing these characteristics as relevant in relation to (high) care utilization [12, 36].

**Table 1. Baseline characteristics of the HNHC population (n = 12 602).**

| Patient characteristics | | |
|---|---|---|
| | **n (%)** | **Missing, n (%)** |
| Demographic characteristics | | |
| Sex | | 0 |
| Male | 4495 (35.67) | |
| Female | 8107 (64.33) | |
| Age, mean (SD)[a] | 67.55 (14.80) | 0 |
| Household position | | 0 |
| Child living at home | 141 (1.12) | |
| Single adult | 3773 (29.94) | |
| Partner with children at home | 1515 (12.02) | |
| Partner without children at home | 6245 (49.56) | |
| Single parent | 403 (3.20) | |
| Member of collective household | 371 (2.94) | |
| Other | 154 (1.22) | |
| Age of children living at parental home | | 0 |
| $\leq 12$ | 388 (3.08) | |
| $>12$ | 1752 (13.90) | |
| No children living at home | 10 462 (83.02) | |
| Biomedical characteristics | | |
| Type of chronic condition(s) | | 0 |
| Only physical | 10 060 (79.83) | |
| Only mental | 436 (3.46) | |
| Combination of both | 2106 (16.71) | |
| Number of chronic conditions, mean (SD) | 2.23 (0.93) | 0 |
| Prevalence of 28 chronic conditions | | 0 |
| Chronic alcohol abuse | 163 (1.29) | |
| Endocardial conditions, valvular conditions | 298 (2.36) | |
| Congenital cardiovascular anomaly | 25 (0.20) | |
| HIV/AIDS | 9 (0.07) | |
| Anxiety disorders | 649 (5.15) | |
| Asthma | 2142 (17.00) | |
| Stroke (including TIA) | 986 (7.82) | |
| Chronic obstructive pulmonary disease (COPD) | 2218 (17.60) | |
| Chronic back or neck disorder | 2033 (16.13) | |
| Coronary heart diseases | 1725 (13.69) | |
| Dementia including Alzheimer's | 172 (1.36) | |
| Diabetes mellitus | 4925 (39.08) | |
| Epilepsy | 181 (1.44) | |
| Hearing disorders | 679 (5.39) | |
| Visual disorders | 1694 (13.44) | |
| Heart failure | 659 (5.23) | |
| Heart arrhythmia | 1446 (11.47) | |
| Cancer | 2032 (16.12) | |
| Migraine | 395 (3.13) | |
| Osteoporosis | 737 (5.85) | |
| Burnout | 452 (3.59) | |
| Osteoarthritis | 2360 (18.73) | |

*(Continued)*

**Table 1.** (Continued)

| Patient characteristics | | |
|---|---|---|
| | **n (%)** | **Missing, n (%)** |
| Personality disorders | 120 (0.95) | |
| Rheumatoid arthritis | 433 (3.44) | |
| Schizophrenia | 53 (0.42) | |
| Mood disorders | 1380 (10.95) | |
| Mental retardation | 48 (0.38) | |
| Parkinson's disease | 136 (1.08) | |
| Socioeconomic characteristics | | |
| Housing situation | | 12 (0.10) |
| Owner-occupied | 6777 (53.78) | |
| Rented[b] | 5813 (46.13) | |
| Source of income | | 0 |
| Paid work[c] | 1974 (15.66) | |
| Social welfare or unemployment benefits | 1838 (14.58) | |
| Pension benefits | 8156 (64.72) | |
| Without income[d] | 634 (5.03) | |
| Number of people in a household with an individual income | | 26 (0.21) |
| 1 | 4594 (36.45) | |
| >1 | 7982 (63.34) | |
| Household dependence on social security payments, mean (SD) | 11.63 (25.44) | 346 (2.75) |
| Paid interest over debts, mean (SD) | 48.89 (782.63) | 20 (0.16) |
| Care utilization | | |
| Pharmaceutical costs | | 16 (0.13) |
| ≤€500 | 4773 (37.87) | |
| >€500 and ≤€1500 | 5122 (40.64) | |
| >€1500 | 2691 (21.35) | |
| GP care utilization before baseline, mean (SD) | 29.97 (18.50) | 0 |

[a] For continuous variables, mean (SD) is reported.

[b] Includes members of collective households.

[c] Includes employees, entrepreneurs, and managers.

[d] Includes students with and without individual income.

Demographic characteristics were measured at baseline and included patients' sex, age (in years), household position (child living at home, single adult, partner with children at home, partner without children at home, single parent, member of a collective household, other), and age of children living at parental home (≤12 years, >12 years (i.e. the age that they generally leave elementary school), no children living at home). Biomedical characteristics were also measured at baseline and included patients' chronic disease diagnoses based on GP care use related to the chronic disease in the 1.5 years before baseline, type of chronic condition(s) (only physical, only mental, or a combination of both), and number of chronic conditions (1 to 28). All socioeconomic characteristics, except for source of income, were measured over the year 2014 and included patients' (household) housing situation (owner-occupied, rented), number of people in a household with an individual income (1, >1), household dependence on social security payments as proportion of gross household income (0% to 100%), and paid interest over debts (in euros, excluding mortgage or debts related to renovating personal property). Source of income (paid work, social welfare or unemployment benefits, pension benefits,

without income) was measured at baseline. Care utilization characteristics included GP care utilization on baseline (number of registred GP consultations) and patients' pharmaceutical costs ($\leq$€500, $>$€500 and $\leq$€1500, $>$€1500) which were measured over 2014.

## Data analysis

Data were validated and checked for outliers and missing values. We employed LCA, which is a sophisticated analysis technique to capture heterogeneity in the HNHC population's needs by the smallest number of unobserved homogeneous classes [37]. Furthermore, LCA is a person-oriented analysis technique [37] which aims to identify classes of individuals with similar patterns of, in the current study, (correlated) personal factors relevant to health care utilization. Initially, the LCA was run using all 41 patient characteristics (see Table 1) in order to explore the potential to identify clinically relevant subgroups. Furthermore, the analysis was conducted with a maximum likelihood estimator with robust standard errors (MLR). Missing values were handled by the default option in the Mplus software (version 8.1). To test whether the missing values were completely at random (MCAR), a MCAR Pearson-Chi Square and Likelihood Ratio Chi-Square test ($P < .05$) was computed. Additionally, the number of random starts values was increased several times to prevent problems related to nonconvergence or local maxima.[38] By stepwise increasing the number of classes, starting with a 1-class model, and comparing various statistical indicators and clinical relevance, we decided on the final model. Statistical indicators for model fit included the Akaike Information Criterion (AIC), [39, 40] Bayesian Information Criterion (BIC), [41] bootstrapped likelihood ratio test (BLRT), [42] and entropy score. Lower values on AIC and BIC indicated better model fit; significant p-values on the BLRT showed dominance of the k class model, compared to the k-1 class model. The entropy score gave an indication of classification certainty, using a cutoff score of at least 0.8, indicating high classification certainty [38]. The BIC and BLRT were considered most important in deciding on the best model as these outperform other statistical indicators [43].

Besides statistical indicators, clinical relevance of the model was a key factor, as the model should support daily clinical practice [15]. Also, the size of the classes within the model was taken into account (also reffered to as substantiality) [15]. A model with classes including at least 10% of HNHC population was considered substantial to counterbalance efforts to tailor interventions in daily practice. Although we aimed to maintain the largest variety of patient characteristics, the model was made more parsimonious after identifying a clinically relevant model. Thus, we removed any variables that did not contribute to the division in clinically relevant classes, significantly deteriorated the model fit, and/or were regarded as being of less added value based on internal clinical insight. Patients in each class of the final model were described in terms of the probability of having a given patient characteristic. In line with previous studies using LCA, probabilities of 70% to 100% were considered high, probabilities of 40% to 69% moderate, and probabilities of less than 40% low [44, 45]. The continuous variables were described by their estimated mean (SE). Furthermore, each class was described in their top five of chronic conditions at baseline and mean GP care utilization (i.e. mean number of weighted GP consultations) over the follow-up period.

## Results

### Baseline characteristics

A total of 63 general practices (48.5%) participated. The complete data set included individual-level data from 58 551 chronically ill patients, of whom 12 602 patients (21.5%) met the inclusion criteria for the study (i.e., were considered HNHC). Baseline characteristics of the HNHC

**Table 2. Statistical indicators and relative class sizes for models with increasing numbers of latent classes.**

|  | 1-class model | 2-class model | 3-class model | 4-class model | 5-class model |
|---|---|---|---|---|---|
| Loglikelihood | -183,726.630 | -172,407.886 | -164,350.740 | -159,286.403 | -154,427.535 |
| AIC[a] | 367,493.259 | 344,893.772 | 328,817.480 | 318,726.806 | 309,047.071 |
| BIC[b] | 367,642.092 | 345,183.995 | 329,249.094 | 319,299.810 | 309,761.466 |
| Entropy | n/a | 0.981 | 0.974 | 0.973 | 0.977 |
| BLRT[c] | n/a | $P < .001$ | $P < .001$ | $P < .001$ | $P < .001$ |
| Relative class size | n/a | 86.62/13.38 | 64.51/23.58/11.90 | 39.30/25.51/23.31/11.87 | 38.18/25.31/18.48/9.21/8.82 |

[a]AIC refers to Akaike Information Criterion

[b]BIC refers to Bayesian Information Criterion

[c]BLRT refers to bootstrapped likelihood ratio test

population, including number (%) of missing values per characteristic, are shown in Table 1. Patients' mean (SD) GP care utilization over the follow-up period was 66.9 contacts (33.3).

## Latent class analysis

A 4-class model was considered statistically and clinically superior. The 4-class model had a low value on BIC, a significant BLRT ($P < .001$), high entropy score (0.973), and each class was sufficiently substantial by including at least 10% of the HNHC population (see Table 2). Although the 5-class model was statistically superior to the 4-class model, it included two classes with less than 10% of the HNHC population and resulted in less relevant and distinct classes compared to the 4-class model. More specifically, a 5-class model largely maintained three of the four classes of the 4-class model and subdivided the fourth and smallest class of the 4-class model into two smaller classes which were relatively indistinct from each other.

Table 3 shows the final model, which includes nine of the initially used 41 patient characteristics and the probabilities of having each patient characteristic, given class membership (see also Fig 1). This means that the following variables were excluded in the final LCA due to less statistical relevance: age of children living at parental home, number of chronic conditions, prevalence of 28 chronic conditions, paid interest over debts, GP care utilization on baseline. The MCAR Pearson-Chi Square and Likelihood Ratio Chi-Square test showed that values were missing completely at random ($P < 0.001$). As the entropy score was high, we report the final class counts and proportions for the latent classes that are based on their most likely latent class membership. Class 1 (n = 4953; 39.3%) had a mean (SE) age of 74.5 years (0.10), had a high probability (0.91) of having a partner but no children at home, and a high probability (0.98) of receiving pension benefits. Based on these dominant characteristics, class 1 was named 'older adults living with partner'. Class 2 (n = 3215; 25.5%) had a mean (SE) age of 78.8 years (0.15), had a high probability (0.92) of being single, and a high probability (0.99) of receiving pension benefits. Based on these dominant characteristics, class 2 was named 'older adults living alone'. Class 3 (n = 2938; 23.3%) had a mean (SE) age of 51.0 years (0.24) and had a high probability of having a partner with or without children at home (0.82). In terms of socioeconomic status, members of class 3 had a moderate probability (0.62) of having paid work. Based on these dominant characteristics, class 3 was named 'middle-aged, employed adults with family'. Class 4 (n = 1496; 11.9%) had a mean (SE) age of 52.2 years (0.32). With regard to household position, members of class 4 had a low probability (0.34) of being single and a low probability (0.33) of having a partner but no children at home. In terms of socioeconomic status, members of class 4 had a high probability (0.84) of receiving social welfare or unemployment benefits. Based on these dominant characteristics, class 4 was named 'middle-

**Table 3. Probabilities of having the (categorical) patient characteristic, given class membership, for each class within the final 4-class model.**

| Patient characteristics | Probability (SE) | | | |
|---|---|---|---|---|
| | Class 1 | Class 2 | Class 3 | Class 4 |
| | (n = 4953) | (n = 3215) | (n = 2938) | (n = 1496) |
| Demographic characteristics | | | | |
| Sex | | | | |
| Male | 0.481 (0.01) | 0.202 (0.01) | 0.290 (0.01) | 0.404 (0.01) |
| Female | 0.519 (0.01) | 0.798 (0.01) | 0.710 (0.01) | 0.596 (0.01) |
| Age, mean (SE)[a] | **74.47 (0.10)** | **78.78 (0.15)** | **51.01 (0.24)** | **52.22 (0.32)** |
| Household position | | | | |
| Child living at home | 0.001 (0.00) | 0.000 (0.00) | 0.034 (0.00) | 0.027 (0.00) |
| Single adult | 0.010 (0.00) | **0.917 (0.08)** | 0.092 (0.01) | **0.341 (0.01)** |
| Partner with children at home | 0.034 (0.00) | 0.000 (0.00) | **0.391 (0.01)** | 0.144 (0.01) |
| Partner without children at home | **0.905 (0.01)** | 0.000 (0.00) | **0.424 (0.01)** | **0.332 (0.01)** |
| Single parent | 0.022 (0.00) | 0.002 (0.00) | 0.044 (0.00) | 0.105 (0.01) |
| Member of a collective household | 0.010 (0.00) | 0.080 (0.07) | 0.002 (0.00) | 0.039 (0.01) |
| Other | 0.019 (0.00) | 0.001 (0.00) | 0.013 (0.00) | 0.012 (0.00) |
| Biomedical characteristics | | | | |
| Type of chronic condition | | | | |
| Only physical | 0.891 (0.00) | 0.863 (0.01) | 0.664 (0.01) | 0.610 (0.01) |
| Only mental | 0.008 (0.00) | 0.014 (0.00) | 0.077 (0.01) | 0.085 (0.01) |
| Combination of both | 0.101 (0.00) | 0.123 (0.01) | 0.259 (0.01) | 0.306 (0.01) |
| Socioeconomic characteristics | | | | |
| Housing situation | | | | |
| Owner-occupied | 0.637 (0.01) | 0.343 (0.01) | 0.723 (0.01) | 0.272 (0.01) |
| Rented | 0.363 (0.01) | 0.657 (0.01) | 0.277 (0.01) | 0.728 (0.01) |
| Source of income | | | | |
| Paid work | 0.018 (0.00) | 0.007 (0.00) | **0.621 (0.01)** | 0.047 (0.01) |
| Social welfare or unemployment benefits | 0.002 (0.00) | 0.002 (0.00) | 0.195 (0.01) | **0.844 (0.01)** |
| Pension benefits | **0.981 (0.00)** | **0.990 (0.00)** | 0.000 (0.00) | 0.042 (0.01) |
| Without income | 0.000 (0.00) | 0.001 (0.00) | 0.184 (0.01) | 0.068 (0.01) |
| Number of people with an individual income in a household | | | | |
| 1 | 0.026 (0.00) | 0.969 (0.01) | 0.218 (0.01) | 0.483 (0.01) |
| >1 | 0.974 (0.00) | 0.031 (0.01) | 0.782 (0.01) | 0.517 (0.01) |
| Household dependence | | | | |
| on social security payments, mean (SE)[a] | 1.28 (0.09) | 0.35 (0.06) | 9.28 (0.33) | 75.81 (0.64) |
| Care utilization | | | | |
| Pharmaceutical costs | | | | |
| ≤€500 | 0.353 (0.01) | 0.318 (0.01) | 0.513 (0.01) | 0.340 (0.01) |
| >€500 and ≤€1500 | 0.439 (0.01) | 0.423 (0.01) | 0.349 (0.01) | 0.378 (0.01) |
| >€1500 | 0.208 (0.01) | 0.259 (0.01) | 0.138 (0.01) | 0.282 (0.01) |

[a] For continuous variables, mean (SE) is reported

aged adults with social welfare dependency'. See also S1 File for a description of typical qualitative personas who characterize the four classes.

In terms of the top five chronic conditions per class at baseline (see Fig 2), diabetes mellitus was most common in each of the four classes, with prevalence ranging from 30.5% in class 3 to 43.4% in class 1. The second most prevalent condition differed between osteoarthritis in class 1 (21.7%) and 2 (23.8%), asthma in class 3 (25.3%), and mood disorders in class 4 (23.1%).

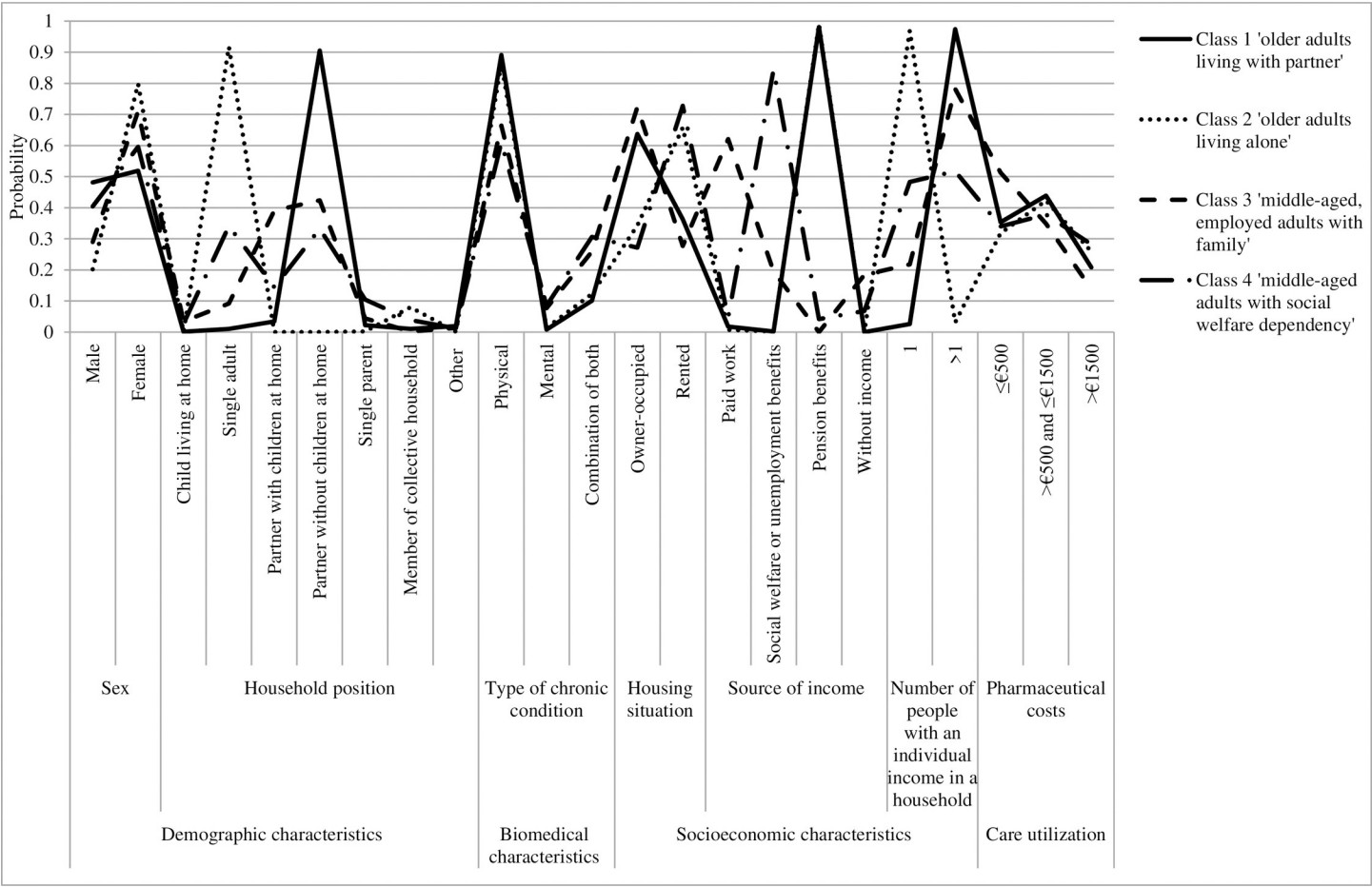

**Fig 1. Probabilities of having the (categorical) patient characteristic, given class membership, for each class within the final 4-class model.**

With regard to GP care utilization of the classes over the follow-up period (see Fig 3), class 2 showed the highest mean care utilization. Both classes with the older adults showed the largest mean (SD) increase in care utilization over time—from 9.8 (6.9) in the first to 11.7 (8.7) in the sixth half year and from 11.5 (8.3) in the first to 14.0 (10.5) in the sixth half year—while the classes with the middle-aged adults were more stable over time—from 10.1 (7.1) in the first to 10.7 (8.2) in the sixth half year and from 11.3 (8.0) in the first to 12.1 (9.5) in the sixth half year.

## Discussion

The present study suggests that the HNHC population in primary care is a heterogeneous population, which can be divided into four subgroups with distinct patterns of particularly demographic and socioeconomic characteristics. Main differences between the subgroups were found in demographic and socioeconomic factors (i.e. age, household position, and source of income). In terms of chronic conditions, the subgroups with older adults most frequently suffered from physical and age-related conditions (e.g. osteoarthritis, cancer), while the middle-aged subgroups most frequently had conditions more typically found in relatively younger people (i.e., asthma and mood disorders). Furthermore, while the subgroups with older adults showed an increase in mean care utilization over time, the middle-aged subgroups showed a

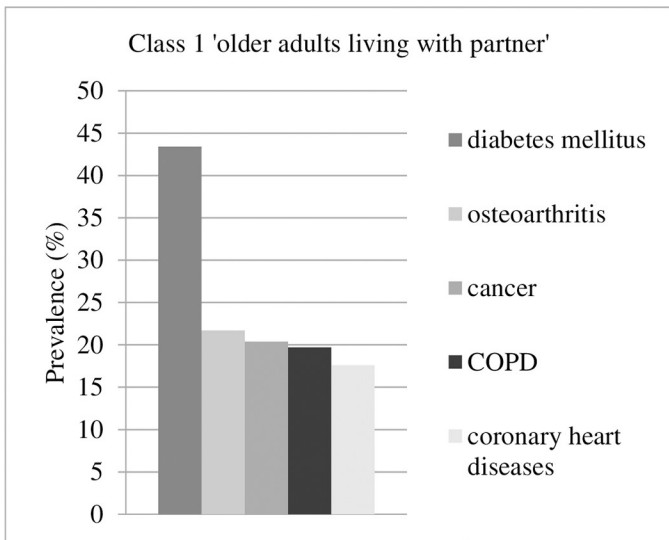

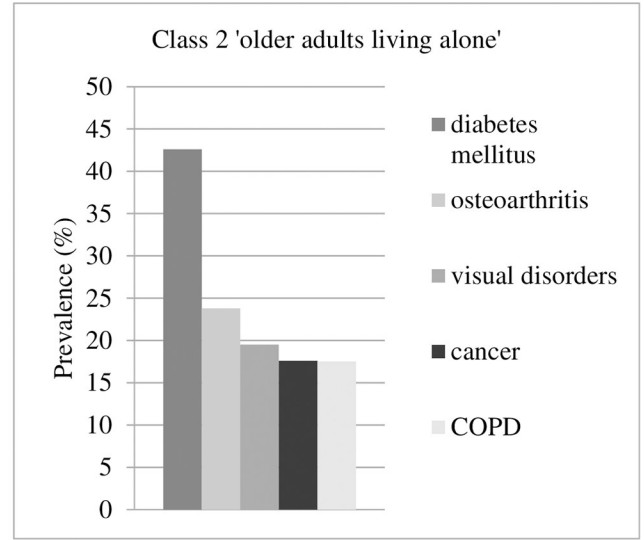

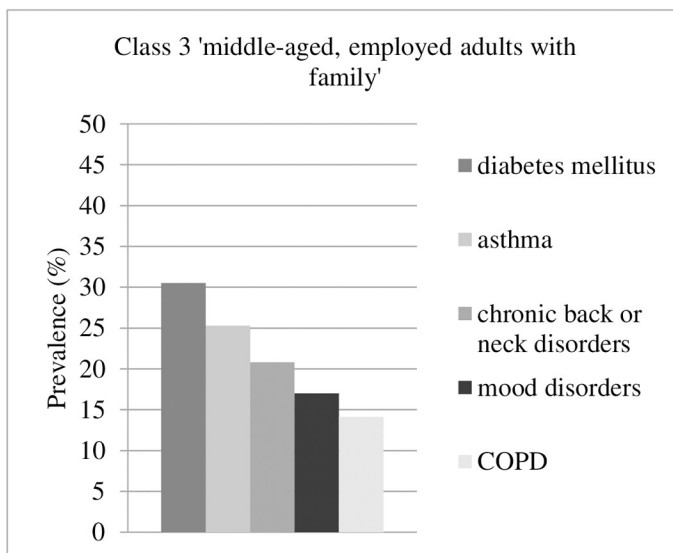

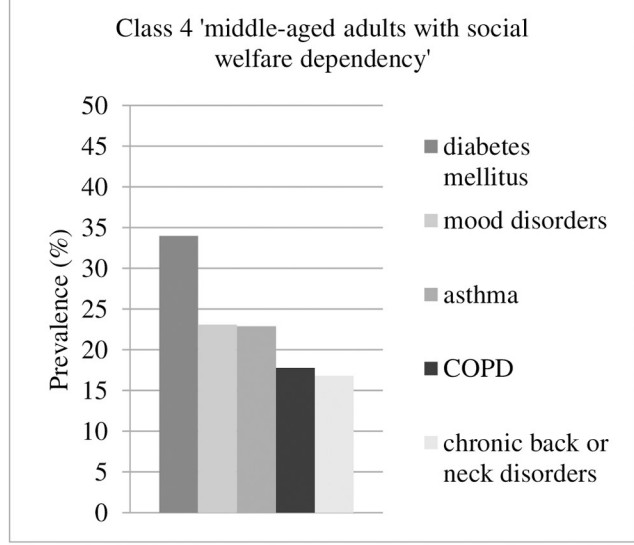

**Fig 2. Top five of chronic conditions (%) per class within the final 4-class model.**

more stable pattern over time. In addition, class 2 ('older adults living alone') showed the highest mean care utilization over time. This finding corresponds with a study of Dreyer, Steventon [36] who showed that living alone is associated with higher care utilization in older adults.

The current study indicates that the sex distribution within the HNHC population, as well as in three of the four identified subgroups, is unbalanced: more than 64% of the HNHC population is female. In the current person-oriented analysis, unlike in a variable-oriented analysis, there is no assessment of relations between variables including corrections for confounders. Rather, the current analysis has focused on identifying subgroups based on patterns of variables within individual patients. One possible explanation for the unbalanced population in terms of sex is that women typically get older and, as a result, are overrepresented among the older aged HNHC patients compared to men. In addition, scientific studies have found that women have significantly higher consultation rates compared to men, but particularly during working years [46, 47].

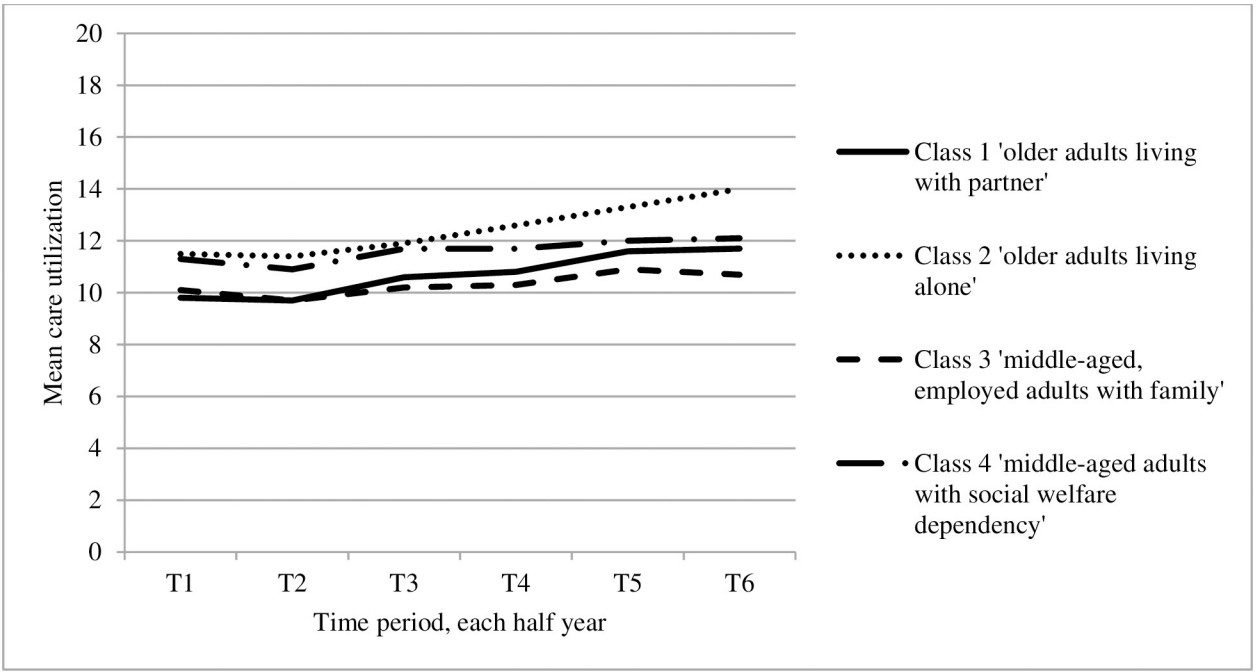

**Fig 3. GP care utilization measured over the follow-up period for each class within the final 4-class model.**

Our findings show that the HNHC population is a demographically and socioeconomically diverse population and includes not only older adults but also many middle-aged people. To date, studies have predominantly focused on (biomedical) segmentation in populations of older adults: an example is the recent Embrace study,[48] which identified three risk profiles for older adults. In line with the demographic heterogeneity found in our study, a study by Wammes, Tanke [12] found that many high-cost patients (in the Dutch curative health system) are not older than 65 years of age. Supporting our approach, the authors [12] emphasized the need for studying the general population with extensive data and targeting interventions toward high-cost patients of various ages. Furthermore, our findings suggest that middle-aged HNHC patients are generally characterized by more socioeconomic vulnerability (e.g., dependence on social welfare) and a higher prevalence of mental conditions (e.g., mood disorders) than are older HNHC patients. These findings add to an increasing awareness about the importance of social and context-related determinants of health [25, 49, 50]. First, Shadmi [51] suggests broadening the understanding and measurement of multimorbidity by including a large variety of health and health-related aspects (e.g., social, cultural, and economic background of populations) that correlate with multimorbidity. In addition, corresponding to our finding that current segmentation often lacks inclusion of relevant demographic and socioeconomic characteristics, the study by Chin-Yee, Subramanian [52] and Khoury, Iademarco [53] also argued that adding environmental and social characteristics (a rather "population perspective") to the genetic profiling in precision medicine can be of added value to public health.

With the growing recognition of the effectiveness of segmentation for patient-centered interventions, [54] the segmentation conducted in the present study can guide clinical practice toward more integrated and person-centered care. By gathering insight into demographic characteristics other than age and gender (e.g., household position) as well as the socioeconomic context of patients (e.g., main source of income), clinical practice in primary care can be attuned to a more holistic view of patients. This view can suggest potentially relevant goals,

interventions, and professionals (within primary care and in cooperation with other disciplines), which can be further discussed in a shared decision-making process with the patient. Such an approach can be inspired by the 'Bridges to Health' model, [55] which aims to systematically connect priority concerns, major components of health care, and goals for health care within identified population segments [56]. Thus, while older adults living alone might benefit from increased social support, middle-aged adults with social welfare dependency might rather benefit from financial and mental support. As such, this segmentation approach can serve as a starting point for more biopsychosocial attention and can inform the discussion of tailored interventions with the patient [56]. However, the individual consultation is still key to assess personal needs and preferences with a patient during a consultation, and agree on an individual treatment course.

Further research, in particular qualitative inquiry, is necessary to identify the most important concerns and components of health care per HNHC subgroup. In addition, the current study has focused on HNHC patients in primary care, which is widely considered the most suitable medical home for chronically ill patients [57]. Although as a result, our findings are mainly useful for improvement of primary care management, there is some evidence that patients with a disproportionately high use of primary care resources also account for significantly high(er) costs in specialist care [58, 59]. For policy making, the subgroups can also help to give insight into the distribution of the patient population over the identified subgroups within certain geographical areas and help to efficiently target resources. In more urban areas, for example, the middle-aged subgroups might be larger than in rural areas.

One of the most important strengths of the current study is the relatively large set of individual-level patient data, with a variety of patient characteristics. A second strength is the use of the model-based analysis technique LCA, which offers a large set of statistical indicators to decide on the best-fitting model and ways to cope with issues of local maxima and nonconvergence [38]. The study also has some limitations. First, individual level data of the non-participating practices were not available in this study. This hampered a direct comparison of participating practices (n = 63; 48.5% of care group) with non-participating practices in order to assess representativeness of the sample. However, particular patient characteristics (i.e. sex, age, household position, and source of income) of the sample were compared to the patient characteristics of the general population in the northern region of the Netherlands that is covered by the primary care group. This comparison showed that the sample is largely similar in patient characteristics to the general population. For example, 50.8% of the sample is female; 50.5% of the general population is female, 20.1% of the sample receives pension benefits; 22.1% of the general population receives pension benefits. Second, EHRs typically include incomplete registrations and may have limited data quality. Nevertheless, the quality of registrations was checked and validated, and the (categorical) missing values were found to be MCAR. Third, the data set included patients who can be considered dependent, as they belonged to the same household. A sensitivity analysis with only completely independent observations showed the same division among classes, implying a negligible effect of the dependent observations on the identification of subgroups. Fourth, only patients with a full EHR registration over the research period were included. This has excluded specific types of patients, such as patients who died before the end of the follow-up period. It is possible that the excluded patients would have been identified as a separate 'near end of life' HNHC subgroup, as identified by some previous population segmentation studies as well [24, 55, 60]. Nevertheless, specific payments arrangements are already in place in Dutch primary care for this patient population who is near the end of life and needs (expensive) palliative care. Fifth, generalizability of the subgroups may be limited, as the data set was retrieved from a specific Dutch region with limited ethnic/cultural diversity and a relatively aged population, compared

to the Dutch average. In future research, the generalizability of the subgroups needs to be determined.

## Conclusions

Despite the heterogeneity of the HNHC population, distinct subgroups with relatively homogeneous patterns of particularly demographic and socioeconomic characteristics can be identified. This study adds to the increasing awareness of the demographic and socioeconomic heterogeneity of the HNHC population, in addition to biomedical diversity. To accommodate person-centered, integrated care delivery, the identified classes need to be connected to tailored care (i.e. concerns, components, goals). This connection can be inspired by the proposed strategies within the 'Bridges to Health' model [55].

## Supporting information

**S1 File. Description of typical qualitative personas who characterize the four identified classes.**
(DOCX)

## Acknowledgments

We thank primary care group 'HZD' and connected general practices for facilitating this study by providing individual-level patient data. Furthermore, we thank all organizations that provided support with either provision, preparation, pseudonymization of data and/or connection of EHR data to other datasets (i.e. 'Calculus', 'ZorgTTP', Statistics Netherlands ('CBS'), 'Vektis'). We also thank researchers Dorijn Hertroijs, PhD; Niels Janssen, MSc, and Sebastian Köhler, PhD, for providing additional assistance in data analysis.

## Author Contributions

**Conceptualization:** Rowan G. M. Smeets, Arianne M. J. Elissen, Mariëlle E. A. L. Kroese, Niels Hameleers, Dirk Ruwaard.

**Data curation:** Rowan G. M. Smeets, Niels Hameleers.

**Formal analysis:** Rowan G. M. Smeets, Niels Hameleers.

**Investigation:** Rowan G. M. Smeets, Niels Hameleers.

**Methodology:** Rowan G. M. Smeets, Arianne M. J. Elissen, Niels Hameleers.

**Supervision:** Arianne M. J. Elissen, Mariëlle E. A. L. Kroese, Dirk Ruwaard.

**Writing – original draft:** Rowan G. M. Smeets, Arianne M. J. Elissen.

**Writing – review & editing:** Rowan G. M. Smeets, Arianne M. J. Elissen, Mariëlle E. A. L. Kroese, Niels Hameleers, Dirk Ruwaard.

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
