## [Decision Letter · Decision Letter 0]

24 Oct 2019

PONE-D-19-25812

Identifying subgroups of high-need, high-cost, chronically ill patients in primary care: A latent class analysis

PLOS ONE

Dear Miss Smeets,

Thank you for submitting your manuscript to PLOS ONE. After careful consideration, we feel that it has merit but does not fully meet PLOS ONE’s publication criteria as it currently stands. Therefore, we invite you to submit a revised version of the manuscript that addresses the points raised during the review process.

The manuscript has been assessed by two reviewers. Their comments are appended below. The reviewers have raised significant concerns about the manuscript.

We would appreciate receiving your revised manuscript by Dec 08 2019 11:59PM. To enhance the reproducibility of your results, we recommend that if applicable you deposit your laboratory protocols in protocols.io, where a protocol can be assigned its own identifier (DOI) such that it can be cited independently in the future. For instructions see: http://journals.plos.org/plosone/s/submission-guidelines#loc-laboratory-protocols

We look forward to receiving your revised manuscript.

Kind regards,

Bruno Pereira Nunes, Ph.D.

Academic Editor

PLOS ONE

Journal Requirements:

'The current study was funded by primary care group 'HZD', the Netherlands (https://www.hzd.nu). The funding agency had no role in study design, data collection and analysis, decision to publish, or preparation of the manuscript.'

We note that you received funding from a commercial source: HZD.

Additional Editor Comments (if provided):

Reviewers' comments:

Reviewer's Responses to Questions

**Comments to the Author**

1. Is the manuscript technically sound, and do the data support the conclusions?

Reviewer #1: Yes

Reviewer #2: Yes

2. Has the statistical analysis been performed appropriately and rigorously? 

Reviewer #1: Yes

Reviewer #2: Yes

3. Have the authors made all data underlying the findings in their manuscript fully available?

Reviewer #1: Yes

Reviewer #2: No

4. Is the manuscript presented in an intelligible fashion and written in standard English?

Reviewer #1: Yes

Reviewer #2: Yes

5. Review Comments to the Author

Reviewer #1: This paper addresses an important issue for health care systems in high income countries. It is analytically sophisticated and might be more consumable if the authors provided typical qualitative personas who might characterize the 4 classes that they identify. One also wonders about the heterogeneity within classes and whether a clinically meaningful approach to this segmentation process would require further breakdown within the 4 groups that emerge statistically in the analysis. The authors should make clear (as noted below) that segmentation suggests but does not establish clinically meaningful interventions.

132: are primary care visits associated with other incurred costs, or is this study useful only to improving management in the primary care setting.

285 ff: authors should be clear that presence of characteristics does not clarify the causal linkages with high utilization or the value of suggested interventions.

Reviewer #2: General comments

The paper offers a straightforward analysis of great relevance for clinical practice and for public policy. What seems like a simple exercise at first – the classification of high-needs, high costs patients (HNHC) into subgroups – resulted in a rigorous, yet interesting paper. On their own, these results may provide important insights and nuance into the understanding of HNHC, for example, that not all these patients are elderly. But they may also be an important starting point for additional qualitative work that could result into useful clinical guidelines. The paper was well-written and I enjoyed reading it. Having said that, some specific points in the methodology and analysis need revisions in order to improve the credibility of the study and strengths of the argument.

Data sources

• 104-105: Practices were invited to provide extract and provide individual-level data, and 48.5% of them responded. It would be important to add a comparison of the characteristics of the respondents and non-respondents. Is it likely that there is any bias in the pattern of responses that would affect the results?

• 108-109: Were the socioeconomic data individually linked to socioeconomic data? Please clarify.

Participants

• 131-132: Even though the study places considerable emphasis on high-cost patients, there is not data on costs generated by this group. Is there any cost data available? Otherwise, I strongly recommend re-labelling the group to high-needs, high-utilisation.

Data analysis

• 180-182: The process for elimination of variables is not transparent and could be interpreted as arbitrary. It would have been good to provide a list of variables that were not considered of clinical significance or had lower added clinical value.

Latent class analysis

• 202-203: While the cut-off of a minimum of 10% established by the authors for each class was clearly spell-out, it is very unlikely that this was a "a priori" design decision. Therefore, the decision to use 4 instead of 5 classes is not very well substantiated. It would be useful to see a comparison of a 5-class model. One way to do this would be to replace the current 1, which does not add any relevant information when compared to the tables for a comparative figure between a 4 and a 5 class model. Given that the main purpose of the paper is to conduct a classification, the argument needs to be stronger.

• 213-214: Are there any other possible living arrangements in class 2? Is there any variable for household size to confirm that these people were indeed living alone?

Table 1 and choice of variables

• The gender distribution of the sample is biased towards females. Why is that? Are females more likely to be high-needs, high-cost patients than men, controlling for age? Some discussion about that should be added to the text. Is it a problem of biased sample?

• How was the cut-off for children living at home decided? Is there any distinction between children and grandchildren in the data?

• For the prevalence of 28 chronic conditions, what is the criteria for inclusion? Are all of these diagnostics current? Do they include past diagnosis, and if so, for how long?

• While several proxies were used for socioeconomic status, some of the most common ones were not used, namely education and income levels. Within the selected variables, there could still be substantial variation. Why weren't education and income used?

6. PLOS authors have the option to publish the peer review history of their article (what does this mean?). If published, this will include your full peer review and any attached files.

Reviewer #1: No

Reviewer #2: No

---

## [Author Response · Author response to Decision Letter 0]

29 Nov 2019

Dear Dr. Heber, Maastricht, November 26, 2019

We are very grateful to be offered the opportunity to revise and resubmit our manuscript entitled: “Identifying subgroups of high-need, high-cost, chronically ill patients in primary care: A latent class analysis” for publication in PLoS ONE. 

We appreciate the helpful feedback that was provided and believe that these comments were relevant and useful to improve our manuscript. Please find below our responses to the comments and reference to the changes we made in the manuscript. The changes in the revised manuscript are shown by ‘Track Changes’.

As requested, we would like to expand the ‘Competing interests’ statement: “AE is currently serving as an Academic Editor for PLOS ONE. Furthermore, our funding source (i.e. primary care group ‘HZD’) is a commercial source. This does not alter our adherence to PLOS ONE policies on sharing data and materials.” 

We sincerely think that we have improved the quality of our paper and we hope that you now consider it for publication in PLoS ONE. 

With kind regards, also on behalf of the co-authors,

Rowan Smeets, MSc

Maastricht University

Faculty of Health, Medicine and Life Sciences

Care and Public Health Research Institute (CAPHRI)

Department of Health Services Research

P.O. Box 616

6200 MD Maastricht, the Netherlands

T: +31(0)43 3881711 / F: +31(0)43 3884162

rowan.smeets@maastrichtuniversity.nl

Response to comments of reviewer #1

We thank the reviewer for the compliments and all the valuable comments provided on the original version of the manuscript. Below we indicate how we handled these comments. In the revised manuscript, all changes to the text are shown by ‘Track Changes’.

 Comment: This paper addresses an important issue for health care systems in high income countries. It is analytically sophisticated and might be more consumable if the authors provided typical qualitative personas who might characterize the 4 classes that they identify. One also wonders about the heterogeneity within classes and whether a clinically meaningful approach to this segmentation process would require further breakdown within the 4 groups that emerge statistically in the analysis. The authors should make clear (as noted below) that segmentation suggests but does not establish clinically meaningful interventions.

Our response: As suggested by the reviewer, we added typical qualitative personas who characterize the four classes as supporting information (with reference to this supporting information, S1 File, in the revised results section) to make the classes more consumable. These qualitative personas include information about the most distinct patient characteristics between the classes (i.e. age, household position, source of income). In addition, information about highly prevalent chronic conditions was included in the personas. For example, the following persona of class 2 ‘older adults living alone’ was provided: “Mrs. Williams is 79 years old and living alone. Her husband has passed away five years ago. For some time now, Mrs. Williams has to deal with visual disorders and osteoarthritis. In addition, she has been suffering from diabetes for a long time.” The segmentation approach presented in the current study is particularly of clinical relevance as it can guide towards (rather than establish) more tailored interventions, in close interaction with the patient. To describe this, we have added the following to the revised discussion: “As such, this segmentation approach can serve as a starting point for more biopsychosocial attention and can inform the discussion of tailored interventions with the patient (56). However, the individual consultation is still key to assess personal needs and preferences with a patient during a consultation, and agree on an individual treatment course.” (page 21, lines 316-320)

Comment: 132: are primary care visits associated with other incurred costs, or is this study useful only to improving management in the primary care setting.

Our response: Our aim was to identify classes of chronically ill patients who can be considered HNHC in primary care, based on their utilization of GP consultations. It was beyond of the scope of the current study to investigate to what degree GP consultations are associated with other incurred costs. This means that the insights retrieved from this study most directly inform management in the primary care setting. Nevertheless, we acknowledge that the association between GP consultations and other incurred costs refers to an interesting topic for future scientific research. Therefore, we added the following to the revised discussion: “In addition, the current study has focused on HNHC patients in primary care, which is widely considered the most suitable medical home for chronically ill patients (57). Although as a result, our findings are mainly useful for improvement of primary care management, there is some evidence that patients with a disproportionately high use of primary care resources also account for significantly high(er) costs in specialist care” (58, 59) (pages 21-22, lines 323-328)

Comment: 285 ff: authors should be clear that presence of characteristics does not clarify the causal linkages with high utilization or the value of suggested interventions.

Our response: We agree with the reviewer that the aim of our analysis should be clearer in the manuscript. Our aim was not to identify relations between a (set of) independent variable(s) and a dependent variable, which are then assumed to hold across all people. Rather than using such a variable-oriented analytic approach, we used an approach more befitting population segmentation purposes, i.e. latent class analysis (LCA). LCA is a person-oriented analytic approach that emphasizes the patient as a whole. LCA aims to identify classes of individuals with similar patterns of personal factors relevant to, in this study, health care use. In selecting personal factors to include in the LCA, we chose a range of 41 characteristics that have been identified as relevant in relation to health care utilization in previous studies. 

To clarify this, the following was added to the revised methods section: “Furthermore, LCA is a person-oriented analysis technique (37) which aims to identify classes of individuals with similar patterns of, in the current study, (correlated) personal factors relevant to health care utilization.” (page 9, lines 160-162) Furthermore, we have added the following to the revised methods section: “Forty-one patient characteristics were initially used as potential indicators of heterogeneity in HNHC patients’ needs in the LCA. These characteristics were included based on scientific studies describing these characteristics as relevant in relation to (high) care utilization (12, 36).” (page 8, lines 137-138) Moreover, we have added the following to the revised discussion: “This view can suggest potentially relevant goals, interventions, and professionals (within primary care and in cooperation with other disciplines), which can be further discussed in a shared decision-making process with the patient. Such an approach can be inspired by ‘The Bridges to Health Model’ (55) which aims to systematically connect priority concerns, major components of health care, and goals for health care within identified population segments (56).” (page 21, lines 309-314) 

Response to comments of reviewer #2

We thank the reviewer for the compliments and all the valuable comments provided on the original version of the manuscript. Below we indicate how we handled these comments. In the revised manuscript, all changes to the text are shown by ‘Track Changes’. 

Comment: 104-105: Practices were invited to provide extract and provide individual-level data, and 48.5% of them responded. It would be important to add a comparison of the characteristics of the respondents and non-respondents. Is it likely that there is any bias in the pattern of responses that would affect the results?

Our response: Although we do not have individual patient data of the non-responding practices, we have compared patient characteristics (i.e. sex, age, household position, and source of income) of responding practices with the general population from this northern region of the Netherlands (that is covered by the primary care group). The (aggregated) data from the general population from this northern region was accessed via the open access database of Statistics Netherlands (i.e. ‘StatLine’). This comparison showed that the participating practices are comparable to the general population in the specific region. To clarify this, we have added the following to the revised discussion: “First, individual level data of the non-participating practices were not available in this study. This hampered a direct comparison of participating practices (n= 63; 48.5% of care group) with non-participating practices in order to assess representativeness of the sample. However, particular patient characteristics (i.e. sex, age, household position, and source of income) of the sample were compared to the patient characteristics of the general population in the northern region of the Netherlands that is covered by the primary care group. This comparison showed that the sample is largely similar in patient characteristics to the general population. For example, 50.8% of the sample is female; 50.5% of the general population is female, 20.1% of the sample receives pension benefits; 22.1% of the general population receives pension benefits.” (page 22, lines 337-346)

Comment: 108-109: Were the socioeconomic data individually linked to socioeconomic data? Please clarify.

Our response: All extracted data were individually linked. To more explicitly describe this, we have added the following to revised methods section: “Furthermore, the EHR data were linked on the individual patient level to socioeconomic data (e.g., source of income) and health care claims data (e.g., pharmaceutical costs).” (page 6, lines 107-108)

Comment: 131-132: Even though the study places considerable emphasis on high-cost patients, there is not data on costs generated by this group. Is there any cost data available? Otherwise, I strongly recommend re-labelling the group to high-needs, high-utilisation.

Our response: We use the terminology of ‘high-need, high-cost’ (HNHC) as the weighting factors that are assigned to the different (primary care) consultation types are related to costs (as we described in the original version of the manuscript under ‘participants’). As such, the population can be considered ‘high-need’ as well as ‘high-cost’, but primarily within our research setting, i.e. primary care. To more explicitly describe this, we added the following to the revised methods section: “As the weighting factors based on time investment are related to costs (35), the patients selected for this study can be considered high-need, high-cost in primary care.” (page 7, lines 133-134)

Comment: 180-182: The process for elimination of variables is not transparent and could be interpreted as arbitrary. It would have been good to provide a list of variables that were not considered of clinical significance or had lower added clinical value.

Our response: To identify clinically relevant subgroups within the HNHC population, we initially employed latent class analysis by using 41 patient characteristics that were considered relevant in relation to (high) care utilization. These 41 characteristics are displayed in table 1. In Table 3, the nine variables included in the final model are shown. This means that 32 variables were excluded (all due to statistical reasons). To clarify this, we added the following to the revised results section: “This means that the following variables were excluded in the final LCA due to less statistical relevance: age of children living at parental home, number of chronic conditions, prevalence of 28 chronic conditions, paid interest over debts, GP care utilization on baseline.” (page 11-12, lines 216-218)

Comment: 202-203: While the cut-off of a minimum of 10% established by the authors for each class was clearly spell-out, it is very unlikely that this was a "a priori" design decision. Therefore, the decision to use 4 instead of 5 classes is not very well substantiated. It would be useful to see a comparison of a 5-class model. One way to do this would be to replace the current 1, which does not add any relevant information when compared to the tables for a comparative figure between a 4 and a 5 class model. Given that the main purpose of the paper is to conduct a classification, the argument needs to be stronger.

Our response: The cut-off of 10% was in fact an important “a priori” decision criterion, because the model also has to be feasible for smaller GP practices that want to provide tailored care, i.e. practices with <2000 patients and <120 HNHC chronically ill patients. Nevertheless, statistical robustness and clinical relevance were firstly assessed and regarded as most important criteria for model choice. To our best knowledge, it is quite uncommon (if the study’s aim is to identify rather than to compare a model) to present a fully detailed (visual) comparison of different models. After all, decision criteria are commonly defined “a priori” and inform model choice. Usually, only a comparison of statistical indicators for models with different number of classes is given (see Table 2 in the original manuscript) and a general substantive argument to underpin the choice of the final model. To make the substantive argument for the choice of the 4-class model stronger, we added the following to the revised results section: “More specifically, a 5-class model largely maintained three of the four classes of the 4-class model and subdivided the fourth and smallest class of the 4-class model into two smaller classes which were relatively indistinct from each other.” (page 11, lines 210-212)

Comment: 213-214: Are there any other possible living arrangements in class 2? Is there any variable for household size to confirm that these people were indeed living alone?

Our response: In class 2, the household position of ‘single adult’ was most dominantly present (0.917), followed by ‘member of a collective household’ (0.08), ‘single parent’ (0.002), and ‘other’ (0.001). There are no other living arrangements in class 2 (see ‘household position’ in Table 3). In our dataset, we have one variable for household size (which was not included in the analysis as it correlates with household position and number of people with an individual income in a household), which confirms that the majority of this class (i.e. 99%) is part of a household with only one member. 

Comment: The gender distribution of the sample is biased towards females. Why is that? Are females more likely to be high-needs, high-cost patients than men, controlling for age? Some discussion about that should be added to the text. Is it a problem of biased sample?

Our response: In the current study, we have used a person-oriented rather than a variable-oriented analysis to seek for subgroups of individuals based on their patterns of patient characteristics. This implies that we did not focus on the relation between (sets of) individual characteristics, with corrections for possibly confounding variables. Whereas the included sample is, compared to the general population, not biased (see response to comment about participating practices), the included HNHC population shows an unbalanced distribution of sex. Several explanations for this unbalanced distribution in sex were added to the revised discussion: “The current study indicates that the sex distribution within the HNHC population, as well as in three of the four identified subgroups, is unbalanced: more than 64% of the HNHC population is female. In the current person-oriented analysis, unlike in a variable-oriented analysis, there is no assessment of relations between variables including corrections for confounders. Rather, the current analysis has focused on identifying subgroups based on patterns of variables within individual patients. One possible explanation for the unbalanced population in terms of sex is that women typically get older and, as a result, are overrepresented among the older aged HNHC patients compared to men. In addition, scientific studies have found that women have significantly higher consultation rates compared to men, but particularly during working years (46, 47).” (pages 19-20, lines 272-281) 

Comment: How was the cut-off for children living at home decided? Is there any distinction between children and grandchildren in the data?

Our response: In response to this comment, we added the following text to the revised methods section: “Age of children living at parental home (≤12 years, >12 years (i.e. the age that they generally leave elementary school), no children living at parental home).” (page 8, lines 142-143) Furthermore, the extracted data (amongst others the variable of ‘Age of children living at home’) only includes information about children (living at home) with regards to household situation; there is no information available on grandchildren (living at home). To make this more clear, we have changed the name of the variable “Age of children living at home” into “Age of children living at parental home”. 

Comment: For the prevalence of 28 chronic conditions, what is the criteria for inclusion? Are all of these diagnostics current? Do they include past diagnosis, and if so, for how long?

Our response: The prevalence of the chronic conditions can be considered the current prevalence/diagnosis as it is based on recent GP care use for the particular condition. To describe this, we added the following to the revised methods section: “Biomedical characteristics were also measured at baseline and included patients’ chronic disease diagnoses based on GP care use related to the chronic disease in the 1.5 years before baseline.” (page 8, lines 143-145)

Comment: While several proxies were used for socioeconomic status, some of the most common ones were not used, namely education and income levels. Within the selected variables, there could still be substantial variation. Why weren't education and income used?

Our response: Educational level was not used as this variable was not available to us on an individual patient level for every (or the majority of) patient(s) covered by the primary care group. Statistics Netherlands only has information on educational level for a sample of the region covered by the primary care group. In addition, individual income was not used (as a continuous variable) as the ‘source of income’ was regarded more informative than the amount of individual income. Therefore, we preferred the inclusion of ‘source of income’ over individual income as a continuous variable.

---

## [Decision Letter · Decision Letter 1]

8 Jan 2020

Identifying subgroups of high-need, high-cost, chronically ill patients in primary care: A latent class analysis

PONE-D-19-25812R1

Dear Dr. Smeets,

We are pleased to inform you that your manuscript has been judged scientifically suitable for publication and will be formally accepted for publication once it complies with all outstanding technical requirements.

With kind regards,

Bruno Pereira Nunes, Ph.D.

Academic Editor

PLOS ONE

Additional Editor Comments (optional):

Reviewers' comments:

Reviewer's Responses to Questions

**Comments to the Author**

1. If the authors have adequately addressed your comments raised in a previous round of review and you feel that this manuscript is now acceptable for publication, you may indicate that here to bypass the “Comments to the Author” section, enter your conflict of interest statement in the “Confidential to Editor” section, and submit your "Accept" recommendation.

Reviewer #1: All comments have been addressed

Reviewer #2: All comments have been addressed

2. Is the manuscript technically sound, and do the data support the conclusions?

Reviewer #1: Yes

Reviewer #2: (No Response)

3. Has the statistical analysis been performed appropriately and rigorously? 

Reviewer #1: Yes

Reviewer #2: (No Response)

4. Have the authors made all data underlying the findings in their manuscript fully available?

Reviewer #1: Yes

Reviewer #2: (No Response)

5. Is the manuscript presented in an intelligible fashion and written in standard English?

Reviewer #1: Yes

Reviewer #2: (No Response)

6. Review Comments to the Author

Reviewer #1: I have no additional comments for the authors or for the editors. The authors have carefully considered and responded to all comments.

Reviewer #2: This is a revised version submitted by the authors in response to a previous round of review. The authors have satisfactorily addressed the comments I raised.

7. PLOS authors have the option to publish the peer review history of their article (what does this mean?). If published, this will include your full peer review and any attached files.

Reviewer #1: No

Reviewer #2: No

---

## [Editor Report · Acceptance letter]

13 Jan 2020

PONE-D-19-25812R1 

Identifying subgroups of high-need, high-cost, chronically ill patients in primary care: A latent class analysis 

Dear Dr. Smeets:

I am pleased to inform you that your manuscript has been deemed suitable for publication in PLOS ONE. Congratulations! Your manuscript is now with our production department. 

With kind regards,

on behalf of

Dr. Bruno Pereira Nunes 

Academic Editor

PLOS ONE